# Running Away from the War in Ukraine: The Impact on Mental Health of Internally Displaced Persons (IDPs) and Refugees in Transit in Poland

**DOI:** 10.3390/ijerph192416439

**Published:** 2022-12-08

**Authors:** Damiano Rizzi, Giulia Ciuffo, Giulia Sandoli, Matteo Mangiagalli, Pietro de Angelis, Gioele Scavuzzo, Mariana Nych, Marta Landoni, Chiara Ionio

**Affiliations:** 1Fondazione Soleterre Strategie di Pace Onlus, 20123 Milan, Italy; 2Unità di Medicina d’Urgenza, Dipartimento di Medicina Interna, Fondazione IRCCS Policlinico San Matteo, 27100 Pavia, Italy; 3CRIdee, Dipartimento di Psicologia, Università Cattolica, 20123 Milan, Italy; 4Dipartimento di Scienze Clinico-Chirurgiche, Diagnostiche e Pediatriche, Unità di Terapia Intensiva, Università degli Studi di Pavia, 27100 Pavia, Italy; 5Fondazione Zaporuka, 03022 Kiev, Ukraine; 6Unità di Ricerca sul Trauma, Dipartimento di Psicologia, Università Cattolica, 20123 Milan, Italy

**Keywords:** psychological support, post-traumatic stress disorder, Ukraine, war trauma, mental health

## Abstract

A growing body of research highlights how communities traumatized by conflict and displacement suffer from long-term mental and psychosocial illnesses. The Russian army’s attack on Ukraine has resulted in an estimated 10 million people being internally or externally displaced from Ukraine, of whom more than 3.8 million have left Ukraine to seek refuge elsewhere in Europe. Soleterre has decided to launch an intervention to provide psychological support to Ukrainian refugees and IDPs, aimed at containing war trauma, assessing the severity of symptoms, and enabling those affected to receive psychological support. The intervention model envisioned the administration of an intake form to provide a rapid collection of qualitative and quantitative information for those arriving in Poland or Lviv from Ukraine. Our results showed how most of the samples reported high or very high levels of anxiety, depression, and sleep disturbances. Moreover, results highlighted how being close to families or being able to keep in touch with them work as a protective factor in enhancing resilience, as well as a support network. These findings underscored the importance of re-thinking our perception of “family” in a broader sense, considering the new facets it can take on in post-conflict situations.

## 1. Introduction

### 1.1. Understanding War Induced Migration: Mental Health and Implications

The armed conflict in eastern Ukraine between the Ukrainian army and pro-Russian separatists has resulted in an estimated 10 million people being internally or externally displaced from Ukraine, of whom more than 3.8 million have left Ukraine to seek refuge elsewhere in Europe [1,2]. It is estimated that the total number of people displaced from Ukraine will increase to about 4 million [3]. Most people seeking protection do so first in countries adjacent to their homeland. More than 2.2 million Ukrainian refugees currently live in Poland. To be near their family and friends, some have traveled to several countries. Evidence shows that the psychological functioning of children and families is significantly affected by war violence [4]. Wars disrupt the social and family networks that promote healthy child development, cause harm and illness, and disrupt the systems that provide preventive, curative, and remedial care [5]. Children and families who relocate abroad due to violence and persecution often face significant psychosocial difficulties along the way, which continue once they arrive in their new environment [4]. Research has repeatedly shown that the circumstances of migration and resettlement can exacerbate or mitigate the effects of armed conflict on mental health. This means that the circumstances refugees experience while travelling to and after arriving in their host countries in addition to the violence and loss caused by conflict have a significant impact on the mental health of refugees and internally displaced persons [6,7]. In practice and around the world, a growing consensus has emerged on the best practices for addressing the diverse psychological and psychosocial needs of people affected by humanitarian disasters [8]. In 2007, a working group of 27 humanitarian organizations, including the International Federation of Red Cross and Crescent Societies, United Nations agencies, and international nongovernmental organizations representing stakeholders from various sectors (health, protection, nutrition, education, water, and sanitation), published guidelines for humanitarian practice that have been widely adopted [9]. According to these recommendations, “mental health and psychosocial support” (MHPSS) refers to “any type of local or external assistance aimed at ensuring or improving psychosocial well-being and/or preventing or treating mental illness” [9]. According to recent studies, major depression and post-traumatic stress disorder (PTSD) are common and persistent among refugees and displaced persons. The highest rates of mental illness are observed in post-conflict countries, where the general population suffers high levels of post-traumatic stress disorder. People in disaster areas suffer much higher levels of psychological distress, including despair, somatization, and anxiety [2]. According to a systematic review of prevalence studies conducted among IDPs (all based on clinical interviews and validated diagnostic systems), 31.5% of respondents suffered from post-traumatic stress disorder (PTSD), 31.5% from depressive disorder, 11.1% from an anxiety disorder, and 1.5% from psychosis. The rates for PTSD and depression are significantly higher than in the general population.

Until recently, the importance of mental health in post-conflict reconstruction was underestimated. Most aid organizations believed in the “rubber band” approach to mental health, which assumed that people would recover and resume their usual lives if they received only basic care [10]. To this end, it is critical to understand the context and realities of war-related flight, migration, and resettlement and the resulting impact on the well-being of these children and families in flight. The value of research that focuses on describing prevalence rates of mental disorders in conflict situations has been questioned. Instead, it is important to identify contextually relevant risk and protective factors at multiple levels of the social ecology, including those at the individual, family, and societal levels [11,12].

### 1.2. A Family Approach: The Importance of Family as a Protective Factor to Enhance Resilience

It is important to understand how migration and conflict affect not only individuals, but also entire families. Families are often uprooted and separated from their homes, and the resulting psychological stress can make it difficult to care for and raise young children [13]. In addition, mental health problems that affect a family’s ability to function may persist long after the conflict has subsided. In order to build a “protective shield” that fully mitigates the impact of socio ecological shocks such as those presented by war and war-related migration, it is essential to analyze the protective capacities and deficits in the systems that surround war-affected children and families [14]. In identifying the various risk and protective variables present in the different supportive layers or social ecologies that surround children and families, a family-based approach can be very useful.

Adversity can become a source of strength when family members are present because they help create a meaningful universe. As a result, family can serve as an anchor for identification and emotional stability [15]. Indeed, the most important goals for providers of mental health care and primary care for refugee and IDPs’ families can be seen as promoting resilience, post-traumatic growth, and strengthening protective factors for refugee children and their families. Although researchers acknowledge the harmful effects of acute stress, in recent years they have begun to take a resilience approach [16,17]. These studies have emphasised that trauma experiences lead to dysfunction, although families survive, recover, and even thrive after negative events [18]. Ungar [19] describes resilience as: “*in the context of exposure to significant adversity, whether psychological, environmental, or both, resilience is both the capacity of individuals to navigate their way to health-sustaining resources, including opportunities to experience feelings of well-being, and a condition of the individual’s family, community and culture to provide these health resources and experiences in culturally meaningful ways*. (p. 225)”. The ability of families and children to “bounce back” and “cope effectively” in the face of extreme hardship and individual, family, and structural stress has been highlighted in the growing literature on resilience and conflict [20]. Family resilience was described by Walsh [21,22] as a dynamic process that uses the relational resources of the family unit to make constructive adjustments that mutually support family members in a difficult situation. However, it is crucial to (re)examine the idea of family. Indeed, this is crucial when thinking about people affected by war, as they may have lost or been alienated from their entire family (as is the case with Ukrainian refugees or internally displaced persons who are women or children only due to mandatory military conscription). Alternatively, they may have formed new family structures based on post-war and post-migration circumstances rather than traditional lineage and kinship patterns [23].

As shown above, trauma and resilience caused by war and armed conflict are family experiences and should be treated as such in discussions and interventions. Despite these facts, the methods and interventions for refugee and IDPs groups devastated by conflict have often focused on “mothers” or “children” in isolation, placing the individual above the family or society as a whole [24].

The influence of multiple interconnected aspects at each level of refugees’ and IDPs’ socio-ecological system has been highlighted, be it the family, peer group, or surrounding culture [20,25]. This was developed to consider socioecological vulnerability and resilience, particularly in relation to understanding the capacities of socioecological systems to minimise vulnerability, absorb significant shocks, and develop adaptive capacity [26].

In this context, the present study aims to investigate the mental health of refugees and IDPs who were in transit from Ukraine to Europe. Due to their legal situation and the fact that they were still in “transit”, this group was considered a particularly interesting research sample. However, due to the many practical challenges associated with studying refugees and IDPs on the move, little is known about the impact of this situation. The specific aims of the study were:1.Explore the mental health and well-being of Ukranian refugees and IDPs who experienced transit conditions;2.Examine the association of mental health and well-being with refugees’ and IDPs’ transit conditions;3.Explore the protective role of family in building resilience by being open and (re)examining the idea of family. Considering the literature on family and war, we focus on the “family” of refugees and internally displaced persons, which may be linked to the biological family or to new members considered as family.

## 2. Materials and Methods

### 2.1. Participants

The study was conducted in an emergency situation, so the sample size was not predetermined. Participants were given the opportunity to decline participation in the study, which was consistent with ethical standards of research practice. They were repeatedly assured that participation in the study would not affect the services they received. 

#### 2.1.1. Sample of Ukrainian Refugees in the Transit Centre in Poland

The sample consisted of 352 Ukrainian refugees (85.5% women and 14.5% men, mean age 45.8 ± 15.7) who had arrived in Poland (in the centre of Przemysl). Most of them reported being married or single (44% and 30.4% of respondents, respectively), while the rest of the sample reported being widowed (10.5%), divorced (9.9%), or in a relationship (2%). The city from which most refugees originated was Kharkiv (24.4%), followed by Kyiv (8.8%) and Zaporizhzhya (8.5%). On average, refugees stayed in the centre of Przemysl for 2.07 days (SD = 2.2) and took 2.66 days to reach it (SD = 3.8). However, 1.2% of the sample reported travelling between 25 and 40 days to reach Poland. Most of them (83%) reported travelling with family and/or friends, while 17% travelled alone. When asked if they would like to return to Ukraine, 2.6% of refugees said yes, whereas most of them (83%) said that they would like to go to other European cities (e.g., Germany (18%), Poland (11.3%), and Denmark (8.2%)), and 11.9% of them did not know where they would go. Most of the refugees (71%) also stated that they had relatives who stayed in Ukraine. On average, only 0.9 (SD = 2.7) days had passed since their last communication with them.

#### 2.1.2. Sample of Ukranian IDP’s in the Transit Centre in Lviv

The sample consisted of 271 Ukrainian IDPs (72.2% women and 27.8% men, mean age 24.2 ± 15.3) in Lviv. Most of them (76.6%) reported being married (38.3%) or accompanied by parents/guardians, 19.1% reported being single, 2.6% were divorced, and the remaining 1.8% were widowed (0.9%) or in a relationship (0.9%). The city from which most refugees originated was Lviv (20.4%), followed by Kyiv (18.2%) and Kharkiv (10.4%). On average, IDPs stayed in the centre for 17.61 days (SD = 18.0) and took 35.6 days to reach it (SD = 42.9). Most of them (94.7%) reported travelling with family and/or friends, while only 5.3% travelled alone. Most of the refugees (49.2%) said that they did not know where they were going to go. When asked if they would like to return to Ukraine, 7.1% said yes, while the remaining 43.7% indicated that they would like to go to other destinations (e.g., Poland (10.7%), Germany (0.7%), and Great Britain (0.2%)). Most of the IDPs (89.3%) also stated that they had relatives who remained in Ukraine and that on average 13.7 (SD = 23.0) days had passed since their last communication with them.

### 2.2. Procedures

The study was conducted in collaboration with the Soleterre Foundation. The Soleterre Foundation is a non-profit foundation and non-governmental organization (NGO). Since 2003, Soleterre has provided health support to the National Cancer Institute and the Institute of Neurosurgery in Kiev. After the Russian army attacked Ukraine, the Soleterre Foundation took immediate action to provide life-saving treatment to young cancer patients who were forced to leave their hospital rooms and seek refuge in bunkers due to the numerous bombardments in Kiev.

Staff from local partner organisations recruited participants while working with the local refugee community to develop safety and psychosocial programmes. For example, the Soleterre Foundation built a listening and psychological support network in three operational scenarios (Ukraine, Poland, and Italy) by activating 20 psychologists who worked remotely, and 18 psychologists and three psychiatrists who worked on the ground in the cities with the most refugees. By building a network with other NGOs on the Polish border, it was possible to track most refugees and IDPs. Staff met with refugees individually and in groups to explain the objectives of the study and to collect the names and contact information of those who were interested in participating. Due to legal constraints on the circumstances, only verbal consent was obtained to participate. Prior to data collection, the lead author briefed staff on interview and assessment techniques, informed consent, and research ethics. To ensure the quality and clarity of the Ukrainian translation and to promote open-ended interviewing tactics, the training included the study and practice of interview questions and assessments.

The research team collected demographic information about the family before each joint interview and entered the information into an interview feedback form (IFF) for those who arrived in Poland (in the center of Przemysl) or Lviv from Ukraine. Respondents were first asked to provide sociodemographic information. At the camps, local staff conducted all interviews and assessments in Ukrainian. Interviewers had no prior relationships with participants and were familiar only with the local Ukrainian refugee community.

Interviews were recorded by staff and then transcribed for the open-ended questions, or completed immediately for the mental health assessment. Translation of the interview was performed by a qualified translator, and the accuracy of the translation was confirmed by the back-translation method [27]. To provide feedback and highlight emerging themes, contradictions, or misunderstandings, as well as to identify opportunities for additional clarification or research, the lead author took field notes and conducted debriefings with interviewees. The main goal of the clinical fieldwork was to identify the most vulnerable individuals and provide them with the opportunity for targeted and individualised psychological support through a network of Ukrainian-speaking professionals.

For the purpose of this study, the measurement used was translated from the original form. The translation into Ukrainian was performed by a qualified translator, and the accuracy of the translation was confirmed by the back-translation method [27].

### 2.3. Measures

For this assessment, a part of the model developed during the emergency COVID-19 at Policlinico San Matteo in Pavia and adapted to war conditions was used [28].

To assess mental health, a revised version of the DSM 5 TR Rated Level 1 Cross Cutting Symptoms was used. The 23 items of the CCSM (APA, 2013) assess 13 mental health domains, including depression, anger, mania, anxiety, somatic symptoms, suicidal ideation, psychosis, sleep disturbance, memory problems, repetitive thoughts and behaviors, dissociation, personality functioning, and substance use. On a scale of 0 (no or mild symptoms) to 4 (very severe or almost daily), respondents rated their experiences in the past two weeks. The instrument was shown to have high psychometric qualities [29]. Given the short time span between the traumatic events of war and the ongoing stressors of transit and detachment, a short version of the instrument was used for the present study, focusing only on four single-item dimensions, namely depression, anger, anxiety, and sleep disturbances to avoid potential triggers and de-escalation of mental health behaviors. IDPs and refugees were asked to indicate the extent of their depression, anger, anxiety, and sleep disturbances on a Likert scale of 0 to 4 (none (0), mild (1), moderate (2), severe (3), or very severe (4)), which was adapted and simplified from the DSM-5 Self-Rated Level 1 transversal Symptom Measure-Adult.

A program of remote or on-site psychological support was then initiated for all participants. For refugees in the transit center in Poland, in-person psychological support was provided for those who would reach Italy, and remote support for those who would go to other countries. To deepen our understanding of family resilience, we also asked sociodemographic questions and open-ended questions about key resilience and war displacement issues. For this analysis, we focus on two main questions: what helps you feel better during this traumatic time and what worries you the most?

### 2.4. Data Analysis 

IBM Spss Statistics version 27.0 (Italian version) was used to analyze the data.

Descriptive analyses were performed for the DSM 5 TR Rated Level 1 Cross Cutting Symptoms. Correlations were performed in the two samples to assess the possible association between the mental health dimensions investigated and some variables related to refugees’ and IDPs’ transit conditions. Finally, t-tests for independent samples and Chi-square tests were used to compare the two samples with respect to the mental health dimensions investigated.

Concerning the qualitative data, to determine temporal data saturation, data were sorted by their receival date, starting with the earliest responses. Data collection was terminated when analysis revealed that the most recent responses did not raise brand new issues, but rather echoed issues that had been highlighted in the earlier responses. It was decided that qualitative content analysis was the best method to shed light on the war phenomenon, as it had not been studied before for the Ukrainian population and lacked a solid understanding. This approach allows a methodical categorization process to identify the content of data, resulting in explicit themes that can be interpreted.

The three phases of the ordering process were open coding, categorization, and abstraction. During the ordering phase, a researcher dove deep into the data and attempted to derive new themes from the collected data. Two impartial raters then gave names to the new categories that contained the group’s themes. Each transcript was read several times during the coding process before being analysed to gain a deeper understanding of the research participants’ experiences and emotions. The categorization process then led to the organisation of the codes. In order to reduce the number of overarching categories, the relevant codes were grouped into larger clusters and the data was then classified. The content analysis process ended with the presentation of the results.

Two impartial raters (ML and GM) coded this subject independently and coded the text content separately before comparing their coding. When different categories were found, they were compared, and in most cases the coders either reached consensus or merged the categories into a more general category. In cases where consensus could not be reached, a third more experienced researcher (CI) was brought in to resolve the discrepancies. The number of agreements among the independent raters, divided by the total number of individually coded items, yielded an inter-rater reliability of approximately 85%.

## 3. Results

### 3.1. IDPs’ Mental Health and Related Themes

Severe (9.8%) or very severe (16.6%) anger was reported by 26.4% of the sample. In addition, 28.9% of IDPs reported severe anxiety (14.6% very severe and 14.3% severe). Considerable severe (26.5%) or very severe (32.3%) depression also occurred in most of the sample. Finally, 17.1% of the IDPs reported severe sleep disturbances (10.6% severe and 6.5% very severe).

A description of means and SD is provided in Table 1.

All the themes identified during the analysis of the responses to open-ended questions are presented in Table 2 and Table 3. Only 5.4% of refugees could not find something that make them feel better. The most important topics mentioned as reasons for well-being in the IDP sample were leisure activities and proximity to family members. Regarding anticipatory anxiety about the future, the two main themes were anxiety about health and uncertainty regarding their futures.

With respect to the correlational analyses, a significant positive correlation emerged between sleep disturbances and the number of days spent at the centre (r = 0.186, *p* < 0.01). Anger was also found to be positively correlated with the duration of the trip (r = 0.193, *p* < 0.01) and the number of days since their last contact with relatives (r = 0.231, *p* < 0.01). Anxiety was positively correlated with the duration of the trip (r = 0.170, *p* < 0.05), the number of days spent at the centre (r = 0.138, *p* < 0.05), and the number of days since their last contact with relatives (r = 0.155, *p* < 0.05). Lastly, depression was positively correlated with the number of days spent at the centre (r = 0.193; *p* < 0.01) and the number of days since their last contact with relatives (r = 0.175; *p* < 0.05).

### 3.2. Refugees’ Mental Health and Related Themes

The adapted scale indicated severe (21.4%) or very severe (19.1%) anger in 40.5% of the sample. In addition, most refugees had severe (23.3%) or very severe (30.5%) anxiety, and severe (31.3%) or very severe (26%) depression. Finally, 15.2% of the refugees reported severe sleep disturbances (10.8% severe and 4.4% very severe). A description of means and SD is provided in Table 4.

The responses to open-ended questions were analysed and the categories are presented in the tables (Table 2 and Table 3). It was found that 0.9% of refugees could not find something that make them feel better. As with IDPs, the main well-being issues cited by refugees were proximity to family members and religion. As for anticipatory anxiety, the main themes were the uncertainty of the future and the war.

With respect to the correlational analyses, a significant negative correlation emerged between sleep disturbances and the duration of the trip (r = −0.120, *p* < 0.05). Moreover, as emerged for IDPs, anxiety was positively correlated with the duration of the trip (r = 0.117, *p* < 0.05), the number of days spent at the centre (r = 0.159, *p* < 0.01) and the number of days since their last contact with relatives (r = 0.151, *p* < 0.01). No significant correlations emerged for the dimensions of anger and depression.

We then compared mental health dimensions across the two samples using independent samples t-tests. As shown in Table 5, the results showed significant differences between the two groups for anger and anxiety. As can also be noted, IPDs show higher levels of mental health problems with respect of all dimensions, except for depression.

Finally, with respect to the Chi-square tests, significant differences were found between the two groups in the dimensions of anger (χ2 = 20.718, *p*= 0.0003), anxiety (χ2 = 69.225, *p* < 0.001), and sleep disturbances (χ2 = 21.339, *p* = 0.0002). In contrast, no significant differences were found with respect to depression.

## 4. Discussion

Evidence shows that families who relocate abroad due to violence and persecution often face significant psychosocial difficulties along the way, which continue once they arrive in their new environment [4]. However, research suggest that in facing adverse conditions, family can serve as an anchor for identification and emotional stability [15].

This study aimed to explore the mental health and well-being of Ukrainian refugees and IDPs as well as explore the protective role of the family in enhancing resilience. Results of the initial analyses conducted on our sample in the centre of Przemysl in Poland and in Lviv are consistent with findings in the literature indicating the occurrence of a variety of psychological symptoms and syndromes in populations in conflict situations [30]. Indeed, the majority of the sample in both Poland and Lviv reported high or very high levels of anxiety, depression, anger, and sleep disturbances. Moreover, results of the correlational analyses conducted highlighted that their mental health conditions were mostly positively related to the duration of their tripand the number of days they spent at the centre, as well as the last contact they had with their loved ones.

These findings are consistent with previous studies [6,7] that demonstrated how the conflict’s consequences, as well as the circumstances refugees and IDPs experience while travelling and when they get to the new environment, significantly impact on their mental health. Moreover, the literature suggests that people in disaster areas suffer much higher levels of psychological distress, including despair, somatization, and anxiety [2]. Furthermore, a positive association emerged between anxiety (for both groups) and anger (for IDPs), and the number of days since their last contact with relatives. This further highlights the crucial role played by the family as a resilience-promoting factor.

A further finding from our analysis shows that refugee or IDP status is associated with mental health. Our t-test showed significant differences between the two groups for anger and anxiety. It is also interesting to remark that IDPs showed higher scores in every dimension, except for depression. Moreover, the Chi-square test performed revealed a difference in the two samples also for the dimension of sleep disturbance. However, a recent overview of systematic reviews [31] highlighted how important groups, such as IDPs, are less likely to be considered in systematic reviews. Indeed, evidence on mental health promotion and early intervention after resettlement is largely absent. Our initial findings emphasised how the journey’s destination (to another country rather than to another place of one’s own) can also have an impact on the mental health outcomes and needs to be explored further.

Further research is needed to investigate the long-term effects of these conditions on mental health. 

To further understand their mental health, qualitative questions were asked about risk and protective factors.

Regarding risk factors, it is clear from their words that the war has left them with a sense of uncertainty about the future, the possibility of one day having a home, and having a homeland to return to. Many mentioned that they are worried about the journey they will have to make. They are afraid of having to learn a new language, not being able to find a place to live, or not having enough money to survive. The war has shaken their sense of safety and security. Refugees expressed concern about their own health and that of their family members.

All these risk factors, if left untreated, could exacerbate the condition of previous or current disorders such as PTSD, anxiety, and depression.

The results on protective factors are interesting. Indeed, refugees and IDPs who have experienced horrific violence and war have also been shown to have resilience and coping mechanisms. When faced with conflict-related adversity, they may indeed exhibit significant resources and adaptive capacities [32,33,34].

Although resilience has historically been viewed as a psychological trait, more recently it has been described as a dynamic process that requires the use of internal and external resources to achieve positive outcomes in the face of adversity [35,36,37]. According to Ungar [19], resilience is *“both the ability of individuals to find their way to the psychological, social, cultural, and physical resources that sustain their well-being and their ability to negotiate individually and collectively to ensure that these resources are provided and experienced in culturally meaningful ways”.*

While internal resources were the focus of many resilience studies [38], a growing number of researchers are beginning to consider “social-ecological resilience”, or the value of resources available to and surrounding people [19,39]. This study takes advantage of this broader concept of resilience, which emphasises the use and accessibility of internal and external resources (collectively referred to as “protective factors”).

These elements could guide the actions of mental health and psychosocial support professionals and subsequently lead to effective interventions aimed at strengthening them.

For this reason, this intervention placed a special emphasis on protective factors. According to previous findings [9,15], results of both qualitative and quantitative analyses show how being close to their families or, at least, being able to keep in touch with them work as protective factors for refugees and IDPs in association with their mental health conditions. Alongside the family, refugees and IDPs emphasised the importance of having a support network of friends, volnteers and people encountered at the centres. Indeed, as previously mentioned, conflict situations such as the one described above confront people with the need to rethink the idea of family because of possible loss or separation from loved ones, as well as the new family structures formed post-conflict [23]. Other relevant protective factors identified were, in order: religion, planning for the future, making sense of the event, leisure activities, and feeling in a safe place.

Qualitative research with Syrian refugees has shown that resilience can manifest itself in the form of group coping mechanisms, which include seeking social support, maintaining contact with relatives in Syria, holding religious beliefs, and developing adaptive fatalism [40].

In summary, to prevent, mitigate, or overcome the negative effects of adversity and contribute to life improvement and/or change, internal qualities and external resources can work independently or interactively, intensely or moderately, and separately or in combination [41].

## 5. Conclusions

Our data underscore the importance of the Soleterre-sponsored intervention and provide a useful initial guide for its implementation. The results of our work add knowledge about the resources to be promoted to foster resilience and psychological well-being in these populations. Indeed, promoting resilience and post-traumatic growth must be considered a primary goal for providers of mental health care and primary care for refugees and IDPs. Moreover, our results highlighted the importance of considering the importance of rethinking the idea of family in a broader sense, in light of the new facets it can take on in post-conflict situations. Volunteers, as well as other refugees and IDPs met during and after travelling, can constitute a solid network of support which can in turn act as a resilience-enhancing factor. These people can provide social activities, community support, and a sense of belonging, all of which are critical to refugees’ and IDPs’ resilience.

This study also has some limitations. Firstly, since the intervention was provided in an emergency condition and at a time when the stressors of transit and detachment were still ongoing, it was not possible to carry out a more structured assessment. We could only use a revised version of the DSM 5 TR Rated Level 1 Cross Cutting Symptoms which was less time-consuming and prevented possible triggers for refugees and IDPs. However, it did make it possible to draw important insights that can guide early interventions aimed at the long-term well-being of these populations. Furthermore, our results provide an initial indication of how mental health conditions also depend on refugee or IDP status that deserves further investigation.

Overall, it is critical to consider the condition of refugees and IDPs in transit as a new problem in the field of trauma-related mental illness. The present study could be considered a precursor to future research and interventions for Ukrainian refugees and IDPs in transit to their home country, or another location of their choice.

## Figures and Tables

**Table 1 ijerph-19-16439-t001:** IDPs’ mean and standard deviations of mental health dimensions.

	Min	Max	Mean	SD
Anxiety	0	4	3.17	1.960
Anger	0	4	1.99	1.434
Depression	0	4	2.47	1.317
Sleep disturbances	0	4	2.66	1.177

**Table 2 ijerph-19-16439-t002:** Themes for the question “What makes you feel better?”.

Coded Theme	Description of Theme	Example Quotations	Percentage Cover in Refugee Sample	Percentage Cover in IDP’s Sample
Nuclear Family Unit	Being closed to each other	*“Being with my daughter”, “when I’m near my mom”*	31.8%	13.9%
Support Network	Friendships, romantic relationships, and other support relationships	*“friends”; “the centre’s volunteers; “boyfriend”; “support from people”*	9.7%	2.8%
Religion	Having faith and hope	*“Pray in Ukraine’s victory over Russia”*	13.9%	4.3%
Future	Envisioning and planning the future	*“Knowing I have a plan”; “thinking about the future”*	6%	0%
Feeling safe	Be in a safe place	*“Feeling safe here”; “not being bombed”*	4.3%	0.9%
Leisure activities:	Engaged in leisure activities	*“movies”; “reading books”*	6%	29.5%
Helping/caring	Help and care about other people	*“to help others”; “taking care of relatives”*	6%	1.7%
Meaning-making Communication	Talk about the war and give an explanation about it	*“Explaining to my grandson why we went away”*	7.4%	10.5%

**Table 3 ijerph-19-16439-t003:** Themes for the question “what worries you the most?”.

Coded Theme	Description of Theme	Example Quotations	Percentage Cover in Refugee Sample	Percentage Cover in IDP’s Sample
War	Triggers from war	*“to see the missiles in Lviv”; “to hear sirens, go to the shelter”; “bomb”*	16.5%	6.8%
Health	Health of one’s own or loved ones’	*“son’s health* *”*	8.5%	12.5%
Travel	Uncertainty and condition of travel	*“The journey to Sweden and the accommodation”*	4.3%	0%
Future	Uncertainty about the future	*“I am worried about the future, whether I will find somewhere to go and whether I will find a job”*	26.4%	11.4%
Feelings	Expressing of negative emotions	*“tension”; “anxiety, fears”; “irritability”; “when my mum cries”*	7.4%	0%

**Table 4 ijerph-19-16439-t004:** Refugees’ mean and standard deviations of mental health dimensions.

	Min	Max	Mean	SD
Anxiety	0	4	2.39	1.205
Anger	0	4	1.57	1.454
Depression	0	4	2.64	1.263
Sleep disturbances	0	4	2.52	1.004

**Table 5 ijerph-19-16439-t005:** Mental health dimensions: *t*-test for independent samples.

Mental Health Dimensions	Sample	Mean	Standard Deviation	T	Sign.	gl
Sleep disturbances	Refugees	2.5	1.0	1.592	0.112	513.363
	IDPs	2.6	1.1			
Anger	Refugees	1.5	1.4	3.511	0.000482	562.936
	IDPs	1.9	1.4			
Anxiety	Refugees	2.3	1.2	5.701	0.00000000228	406.671
	IDPs	3.1	1.9			
Depression	Refugees	2.6	1.2	1.620	0.106	547.447
	IDPs	2.4	1.3			

## Data Availability

The dataset in not publicly available to maintain the participants’ privacy.

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
