# Peer review of "Running Away from the War in Ukraine: The Impact on Mental Health of Internally Displaced Persons (IDPs) and Refugees in Transit in Poland"

_ijerph, 2022, doi:10.3390/ijerph192416439_

Round 1
Reviewer 1 Report
The topic of this research is very important and very valuable to have insight into the current psychological issues of IDPs and Refugees from Ukraine.
I was looking forward see the result of this research but was a little disappointed by very basic level of data presentation and how those data have been analyzed.
There is a mixture of quantitative and qualitative data, but neither of them was shown and discussed appropriately.
Author Response
Thank you for taking the time for reviewing our article.
We are grateful for the meaningful suggestion you gave us.
We have followed yours suggestions, trying to answer all comments.
You can find our response for each point highlight below.
Comment #1: The topic of this research is very important and very valuable to have insight into the current psychological issues of IDPs and Refugees from Ukraine.
Reply#1: We thank the reviewer for the positive comment.
Comment #2: I was looking forward see the result of this research but was a little disappointed by very basic level of data presentation and how those data have been analyzed.
Reply#2: we followed your advice and tried to improve our analysis and the presentation of our data.
Comment #3: There is a mixture of quantitative and qualitative data, but neither of them was shown and discussed appropriately.
Reply#3: we followed your advice and improve the quality of our data presentation.
Reviewer 2 Report
The article “Running away from the war in Ukraine: The impact on mental health of Internally Displaced Persons (IDPs) and Refugees in transit in Poland” deals with a relevant and timely topic, and presents relevant and interesting original data.
However, some minor, but also major changes are necessary if the article was to get published.
Generally, the article should pay much more attention to gender, since most Ukrainian refugees (and study participants) are women. Results should be interpreted accordingly and compared to other relevant studies, instead of being gender-blind. The definitions of main concepts used and/or measured are also missing (e.g. resilience, mental health, development) from the introductory part of the article. Methodological part of the article should provide more details as well (see below).
The estimates on the number of Ukrainian refugees (internal and international) mentioned in the Abstract and in the main text (93-96) should be updated according to the latest available sources, since the numbers have risen significantly since March 2022.
Authors should describe in more detail how and why “mental health is increasingly recognized as an important development issue” at the very beginning of the text, to further illustrate the social significance of their article.
The sentence (66-69) should be substantiated with relevant references.
In order to avoid using “countries” twice in the sentence (70-72), I suggest a reformulation (e.g. “…not limited to war-torn areas but spread to entire countries and regions…”).
I am not a psychologist, but the sentence (86-88) seems to lump different emotions, states, and behaviours as examples of “transient emotions”. Are domestic or sexual violence examples of transient emotions?
Since Ukraine is in Europe, it would be more precise to state “search of asylum in other European countries” or “countries of the EU” (22, 94, also similar in line 109 – should be “to other countries in Europe” or something like that).
Authors should further explain what they mean by “numerous practical challenges associated with studying migrants while they are travelling” (112-113).
I would suggest using the term “study participants” or “respondents” instead of “subjects” (149).
I would also suggest using the term “general population” instead of “general public” (162-163).
Data collection procedure (both quantitative and qualitative) must be described in much more detail, as well as data processing and analysis when it comes to qualitative content analysis. How were the answers recorded? Was it all on the same form, with open ended questions? Did the refugees/IDPs fill the form themselves, or somebody recorded their answers? When was the study conducted? Over what time span?
When discussing details and study results on a “Sample of Ukrainian Refugees in the transit centre in Lviv” it seems more precise to use the term IDPs (p. 6), in line with the title of the article. It might seem odd why people originating from Lviv (217-218) are considered refugees in the transit centre in Lviv. To avoid confusion, authors should describe the role and functioning of transit centres where they undertook the research in more detail.
In Conclusions section, authors compare the results of their study to other studies on “migrants” (250-252; 265-266). “Migrants” and “refugees” are not the same terms, nor can refugee (forced) migration be equated with other types of migration. The concept of “resilience” should be explained in the introductory part of the article (see above).
Generally, I would recommend authors to integrate different parts of the article more adequately, and to explain the details, limitations and implications of their study in more depth.
Author Response
Thank you for taking the time for reviewing our article.
We are grateful for the meaningful suggestion you gave us.
We have followed your suggestions, trying to answer all comments.
You can find our response for each point highlight below.
Comment #1: The article “Running away from the war in Ukraine: The impact on mental health of Internally Displaced Persons (IDPs) and Refugees in transit in Poland” deals with a relevant and timely topic, and presents relevant and interesting original data.
Reply#1: We thank the reviewer for the positive comment.
Comment #2: Generally, the article should pay much more attention to gender, since most Ukrainian refugees (and study participants) are women. Results should be interpreted accordingly and compared to other relevant studies, instead of being gender-blind. The definitions of main concepts used and/or measured are also missing (e.g. resilience, mental health, development) from the introductory part of the article. Methodological part of the article should provide more details as well (see below).
Reply#2: we have followed your advice and added to the introduction an explanation of the meaning of gender and deepened the description of the main consructs and methodology
Comment #3: The estimates on the number of Ukrainian refugees (internal and international) mentioned in the Abstract and in the main text (93-96) should be updated according to the latest available sources, since the numbers have risen significantly since March 2022.
Reply#3: we have updated the numbers according to the last sources.
Comment #4: Authors should describe in more detail how and why “mental health is increasingly recognized as an important development issue” at the very beginning of the text, to further illustrate the social significance of their article.
Reply#4 we have followed the advice and added a whole subchapter on the importance of mental health of refugees and IDPs
Comment #5: The sentence (66-69) should be substantiated with relevant references.
Reply#5: In the redesign of the introduction we have deleted the sentence
Comment #6: In order to avoid using “countries” twice in the sentence (70-72), I suggest a reformulation (e.g. “…not limited to war-torn areas but spread to entire countries and regions…”).
Reply#6: we reworded the sentence.
Comment #7: I am not a psychologist, but the sentence (86-88) seems to lump different emotions, states, and behaviours as examples of “transient emotions”. Are domestic or sexual violence examples of transient emotions?
Reply#7: we modified the sentence.
Comment #8: Since Ukraine is in Europe, it would be more precise to state “search of asylum in other European countries” or “countries of the EU” (22, 94, also similar in line 109 – should be “to other countries in Europe” or something like that).
Reply#8: we reworded those sentences.
Comment #9: Authors should further explain what they mean by “numerous practical challenges associated with studying migrants while they are travelling” (112-113).
Reply#9: In the redesign of the introduction we have deleted the sentence
Comment #10: I would suggest using the term “study participants” or “respondents” instead of “subjects” (149).
Reply#10: we replaced the term “subjects”.
Comment #11: I would also suggest using the term “general population” instead of “general public” (162-163).
Reply#11: we replaced the term “public”.
Comment #12: Data collection procedure (both quantitative and qualitative) must be described in much more detail, as well as data processing and analysis when it comes to qualitative content analysis. How were the answers recorded? Was it all on the same form, with open ended questions? Did the refugees/IDPs fill the form themselves, or somebody recorded their answers? When was the study conducted? Over what time span?
Reply#12: We specified better the procedure.
Comment #13: When discussing details and study results on a “Sample of Ukrainian Refugees in the transit centre in Lviv” it seems more precise to use the term IDPs (p. 6), in line with the title of the article. It might seem odd why people originating from Lviv (217-218) are considered refugees in the transit centre in Lviv. To avoid confusion, authors should describe the role and functioning of transit centres where they undertook the research in more detail.
Reply#13: we followed your advice and corrected the term “refugees” in order to avoid confusion.
Comment #14: In Conclusions section, authors compare the results of their study to other studies on “migrants” (250-252; 265-266). “Migrants” and “refugees” are not the same terms, nor can refugee (forced) migration be equated with other types of migration. The concept of “resilience” should be explained in the introductory part of the article (see above).
Reply#14: we rearranged this section as suggested.
Comment #15: Generally, I would recommend authors to integrate different parts of the article more adequately, and to explain the details, limitations and implications of their study in more depth.
Reply#15: we added this information in conclusion section.
Round 2
Reviewer 2 Report
Authors have addressed most of the issues from the first version of the paper successfully. The paper is now more integrated and generally more precise.
The authors have added a sentence in the Abstract and in the Introduction that does not describe the ongoing conflict fully nor accurately – it is not limited to Eastern Ukraine nor are "the Ukrainian army and pro-Russian separatists" only sides in that conflict. Please consider changing the formulation, or removing that sentence completely. Earlier version of the paper described the conflict as "Russian army's attack on Ukraine" which is correct.
Author Response
Dear reviewer 2,
We thank you for the comment. We have deleted the sentence in the abstract and provided a new statement according to your suggestions :
Authors have addressed most of the issues from the first version of the paper successfully. The paper is now more integrated and generally more precise.
The authors have added a sentence in the Abstract and in the Introduction that does not describe the ongoing conflict fully nor accurately – it is not limited to Eastern Ukraine nor are "the Ukrainian army and pro-Russian separatists" only sides in that conflict. Please consider changing the formulation, or removing that sentence completely. Earlier version of the paper described the conflict as "Russian army's attack on Ukraine" which is correct.